# Prospective Comparative Analysis of Simultaneous Microbiological Assessment in Septic Revision Arthroplasty: Can We Rely on Standard Diagnostics?

**DOI:** 10.3390/jcm14155582

**Published:** 2025-08-07

**Authors:** Tobias Freitag, Marius Ludwig, Olivia Trappe, Moritz Oltmanns, Heiko Reichel, Michael Fuchs

**Affiliations:** Department of Orthopaedic Surgery, University of Ulm, Oberer Eselsberg 45, 89081 Ulm, Germany; marius.ludwig@rku.de (M.L.); olivia.trappe@rku.de (O.T.); moritz.oltmanns@rku.de (M.O.); heiko.reichel@rku.de (H.R.); michael.fuchs@rku.de (M.F.)

**Keywords:** periprosthetic joint infection, pathogen, contamination, microbiology, septic revision surgery, microbial diagnostics

## Abstract

**Background:** Microbial analyses of tissue samples are of paramount importance for diagnostic and therapeutic purposes in the course of septic revision arthroplasty. Isolation and identification of the causative pathogens pave the way for successful treatment of periprosthetic joint infections, which necessitates a reliable microbiological workup. It is unknown if there are inconsistencies in pathogen detection and differentiation between accredited laboratories in the context of septic revision arthroplasty. **Methods:** Tissue samples of forty consecutive patients undergoing septic total hip and knee revision surgery were sent to two different accredited and certified laboratories and tested for pathogen growth and bacterial differentiation. **Results:** Each institution analyzed 200 specimens. Twenty-five patients (62.5%) showed consistent results between laboratories. Diverging results were observed in 15 of 40 patients (37.5%). Of these, three individuals showed pathogen growth in only one laboratory. In 12 patients with discrepant results, laboratory analyses revealed a partly different pathogen spectrum. With regard to clinical impact and infection eradication, the respective differences implicated a therapeutic response by a change of the administered postoperative antibiotic treatment in five (12.5%) of the patients. The kappa correlation coefficient indicated a slight value in terms of data consistency between institutions (k = 0.227, *p* = 0.151). **Conclusions:** The majority of evaluated samples show comparable results with regard to microbiological evaluation. Nevertheless, a substantial number of specimens were classified differently. The observed discrepancies pose a challenge for postoperative decision-making. Against this background, standardized microbiological protocols remain mandatory for a conclusive clinical implication to eradicate PJI.

## 1. Introduction

Periprosthetic joint infection (PJI) represents one of the most serious and challenging causes of implant failure after total hip (THA) and knee arthroplasty (TKA). During septic revision surgery, accurate microbiological analysis is crucial, as it significantly impacts subsequent surgical treatment and antimicrobial therapy [1,2]. Despite the critical importance of pathogen identification, the diagnostic accuracy of intraoperatively obtained tissue samples can be affected by several preanalytical and analytical factors [3,4]. These include, among others, prior antibiotic treatment, sampling technique, and sample handling and transport, as well as laboratory processing protocols, all of which can impact the detection and classification of causative microorganisms. Current consensus guidelines recommend collecting multiple tissue samples with consecutive microbial analysis during THA and TKA revision surgery [5,6]. However, the consistency of microbial results, especially in complex cases with low-grade or polymicrobial infections, remains subject to multiple determinants and uncertainties. This is of particular relevance in light of recent findings of our study group, which suggest that interlaboratory differences in microbial analyses during aseptic TKA and THA revision surgery may influence the diagnostic yield as well as the therapeutic approach [7]. In a previous study, we evaluated intraoperative microbiological findings during aseptic revision surgery and demonstrated a significant variability between two internationally certified and accredited laboratories [7]. While these findings have been described in aseptic situations, it remains unclear whether similar diagnostic uncertainties and interlaboratory inconsistencies affect cases of septic revision, in which accurate identification of the causative pathogen is particularly crucial for successful treatment. Due to the inherent question of diagnostic reliability, the aim of this study was to analyze microbiological results from septic revision hip and knee arthroplasty procedures and to compare the diagnostic accuracy between two different laboratories. We hypothesized that in septic situations, interlaboratory results would differ in terms of pathogen identification rate and bacterial species classification. Consequently, the purpose of this study was to estimate the prevalence of diverging laboratory results as well as to point out their clinical implications.

## 2. Materials and Methods

### 2.1. Study Design

Between 1 August 2021, and 3 August 2022, 40 consecutive patients aged 18 years or older who underwent septic revision THA or TKA were prospectively included. The study was approved by the local ethics committee (registration number: FSta 40/20). Written informed consent was obtained from all participants prior to enrollment. Preoperative assessment was conducted in our outpatient department and included clinical examination, systemic blood sampling, radiographic imaging, and joint aspiration. In the presence of severe periarticular bone loss or unclear implant loosening, additional CT imaging was performed. PJI was defined according to the guidelines of the European Bone and Joint Infection Society (EBJIS) [5]. Only patients with confirmed PJI were included for further analysis. Patients who had received antibiotic therapy within 2 weeks before surgery or who were receiving systemic immunosuppressive treatment were excluded from further evaluation.

### 2.2. Revision Surgery and Sample Collection

All revision procedures were performed by 3 senior high-volume orthopedic surgeons. Surgeries were conducted in operating rooms equipped with laminar airflow systems and certified under DIN EN ISO 14644-3 standards [8]. In each case, we collected 5 periprosthetic tissue samples for histopathological evaluation and 5 samples for microbiological analysis, all under a standardized protocol. Histological specimens were harvested from the prosthesis–bone interface and classified according to the Morawietz and Krenn criteria [9]. All samples were placed in a sterile tube without culture medium or saline. Consistent with our previous study on aseptic revisions, a duplicate set of microbiological samples (2 × 5 per patient) was collected to allow simultaneous processing by two independent laboratories. Hence, each institution analyzed 5 tissue specimens for microbiological analysis.

### 2.3. Sample Processing

Microbiological specimens were processed by two laboratories, both accredited according to DIN EN ISO 15189 [10]. Regarding sample acquisition, sample handling, transport, and specimen processing, a previously established standardized protocol was applied [7,11]. Respective specimens were incubated in enrichment broths (Laboratory 1: thioglycollate broth, Thermo Fisher Scientific Inc., Waltham, MA, USA; Laboratory 2: thioglycollate plus brain heart infusion broth, Thermo Fisher Scientific Inc., Waltham, MA, USA) and subsequently plated on 5 different agar media in the presence of pathogen growth (Laboratory 1: Schaedler Agar, Thermo Fisher Scientific Inc., Waltham, MA, USA; CHROMagar Candida, CHROMagar, Saint-Denis, France; Columbia CNA Agar, MacConkey Agar, Columbia Agar, bioMérieux, Marcy l’Etoile, France; Laboratory 2: MacConkey Agar, bioMérieux, Marcy l’Etoile, France; CasoAgar, Chocolate Agar, Schaedler KV Agar, Schaedler Agar, Thermo Fisher Scientific Inc., Waltham, MA, USA). Incubation was performed at 35 °C for 14 days in Laboratory 1 and at 35 °C for 10 days in Laboratory 2. Microbial identification and susceptibility testing were performed using an automated analysis system (Vitek 2, bioMérieux, Marcy-l’Étoile, France).

### 2.4. Rating of Bacterial Results

Microbiological results were classified as follows: Two samples with growth of the same pathogen or detection of a highly virulent pathogen in one sample was classified as periprosthetic infection. Pathogen growth in a single sample was carefully analyzed according to EBJIS-specific definitions (in accordance with preoperative joint aspiration or presence of common contaminants): Intraoperative samples generating a single moderate growth of a low-virulence pathogen were classified as contamination if preoperative joint aspiration remained sterile. In the presence of single moderate growth of a low-virulence pathogen and a simultaneous growth of the same pathogen during preoperative joint aspiration, PJI was confirmed.

### 2.5. Statistical Analysis

Statistical analyses were performed using SPSS software (IBM SPSS Statistics for Windows, version 29.0; IBM Corp, Armonk, NY, USA). Interlaboratory agreement was assessed using Cohen’s kappa (κ) coefficient. Unless otherwise specified, data are presented as the mean ± 1 standard deviation. A 2-sided *p* value < 0.05 was considered statistically significant.

## 3. Results

### 3.1. Demographics and Surgical Procedures

Forty patients were included in this study. The mean age at the time of operation was 79.9 ± 9.2 years (range: 49.0–87.3 years). Fourteen patients (35%) were male. All patients were preoperatively diagnosed with PJI using the above-mentioned diagnostic procedures and criteria [5]. Revision TKA was performed in 18 cases (45%), whereas 22 patients (55%) underwent revision THA procedures.

With regard to revision TKA surgeries, the following surgical procedures were applied: (1) debridement, synovectomy, irrigation, liner exchange, antibiotic therapy, and implant retention according to a debridement, antibiotics, irrigation, and implant retention (DAIR) procedure (14 patients, 77.8%); (2) two-stage septic revision by primary implant removal, synovectomy, irrigation, and temporary antibiotic spacer implantation (4 patients, 22.2%). THA revision surgeries either included a DAIR procedure (8 patients, 36.4%), or an implant removal followed by consecutive articulating antibiotic spacer implantation (14 patients, 63.6%).

### 3.2. Laboratory Results

All in all, 400 microbial specimens were evaluated. Among the 200 samples sent to Laboratory 1, 107 specimens (53.5%) yielded positive cultures, while 120 out of the 200 samples of Laboratory 2 showed bacterial growth (60%).

In 25 patients (62.5%), the results were consistent between the two evaluated laboratories. The kappa correlation coefficient indicated a slight value in terms of data consistency between institutions (k = 0.227, *p* = 0.151, 95% confidence interval [−0.085, 0.539]). Simultaneous bacterial growth of the same pathogen was found in two or more samples per laboratory in 18 of 25 patients. In seven individuals, the respective specimens displayed no pathogen detection and remained sterile. In these cases, PJI was diagnosed by elevated cell count and neutrophil percentage in the presence of an infectious periprosthetic membrane (type II or III according to Morawietz and Krenn) in four cases. One patient showed a type II membrane with normal cell count. In two patients, a PJI was highly suspected due to the combination of history of present illness, clinical examination, and radiological signs of osteolysis. These two patients underwent surgery for tissue biopsy to ensure the diagnosis and presented with a type 3 membrane according to Morawietz and Krenn.

Diverging results between laboratories were observed in 15 patients (37.5%, Table 1). In three cases (7.5%), pathogens were only detected in one laboratory, while all samples in the other institution remained sterile (patients 1, 13, 15). These patients were treated according to the respective antibiogram constellation of the detected pathogen.

In the other 12 patients, partly diverging pathogens were detected. These cases were further differentiated based on whether the respective differences indicated a diverging therapeutic approach in terms of antibiotic treatment.

In 10 of these individuals (patients 2, 3, 4, 6, 7, 8, 9, 11, 12, and 14), the discrepancies led to no change in antibiotic therapy (Table 1). In three patients with polymicrobial findings (patients 2, 3, and 9), a fistula was present. Considering pathogen differentiation and respective antibiograms, the scheduled antibiotic therapy was similar between the two laboratory results. In five patients (patients 4, 6, 7, 8, and 12), a single pathogen identification was classified as contamination (Table 1). The pathogens in patient 14 were labeled differently, while the respective antibiograms revealed similar sensitivity and resistance, resulting in the same antibiotic treatment regimen.

In two patients (patients 5 and 10), the diverging results led to a more extensive antibiotic treatment compared to the result interpretation of one institution.

In patient 5, the detection of Acinetobacter baumanii resulted in additional treatment with Trimethoprim/sulfamethoxazole compared to an isolated treatment with amoxicilline for Cutiacterium acnes. In patient 10, the antibiograms of the detected pathogens differed as Staphylococcus lugdunensis was oxacillin-susceptible while Staphylococcus pseudintermedius was declared oxacillin-resistant. For this reason, antibiotic therapy with fosfomycin was extended by adding daptomycin.

## 4. Discussion

According to current guidelines, intraoperative tissue sampling and subsequent microbial analysis are key steps during THA and TKA revision surgery [5,6,12]. In this context, the acquisition of five intraoperative tissue samples is recommended [5,6,12]. Especially in the course of septic revision arthroplasty, meticulous analysis of culprit pathogens paves the way for a successful PJI treatment.

However, it has been demonstrated that microbial analyses are susceptible to specific contaminations during sample acquisition, handling, and transportation [2,13,14].

In this context, diagnostic standards such as those of the European Bone and Joint Infection Society (EBJIS) and the Infectious Diseases Society of America (IDSA) emphasize the importance of pathogen concordance and quantitative thresholds in defining PJI [5,12]. Yet, these definitions rely heavily on the accuracy of microbiological culture results, which, as our study demonstrates, can be undermined by interlaboratory variability—even when protocols are standardized.

In the presence of pathogen growth, it is furthermore conceivable that different laboratory analyses reveal different subspecies of the same bacterial strain. As such, the question of inherent reliability and consistency of microbiological testing arises. Consequently, this study aimed to analyze microbiological findings from septic revision THA and TKA procedures and to compare the diagnostic accuracy between two different laboratories.

The main finding of this study reveals a slight and relatively low correlation between the observed institutions in terms of data consistency (k = 0.227, *p* = 0.151). More precisely, this means that while there is some agreement, it is not strong and is only slightly better than what might be expected by chance. 

Taken together, our findings indicate that the majority of evaluated samples (62.5%) show comparable results. Nevertheless, it should be noted that a substantial number of patients (*n* = 15) were classified differently, resulting in a rate of 37.5% for discrepant findings. Of these, three (20%) patients (patients 1, 13, and 15) showed pathogen growth in only one laboratory. Patients were treated according to the antibiogram of the detected pathogen. It remains speculative whether these findings are related to differences in sample handling or processing methods. Against this background, it is worth noting that there was a slight difference in incubation time: the latter was performed at 35 °C for 14 days in Laboratory 1 and at 35 °C for 10 days in Laboratory 2. However, respective positive results were displayed within 1 week after index surgery and were not solely associated with one institution.

In 12 (80%) of the 15 patients with discrepant results, laboratory analyses revealed pathogen growth in both institutions, with a partly different pathogen spectrum. Regarding the individual therapeutic approach, the respective differences did not result in a change of the preferred antibiotic classes to treat PJI in 10 of 12 patients. Thus, even in the face of the observed deviations, 10 of 15 patients (66.6%) with discrepant microbiological results would have been treated identically if only one of the respective analyses had been present. In addition to the above-mentioned three patients with single-pathogen growth in just one laboratory, two patients (patients no. 5 and 10) exhibited divergent results, which led to a more extensive treatment in terms of antibiotic class application, which was chosen to be of wide-spectrum coverage.

The consequences of incorrect pathogen identification may imply significant patient-specific complications: False-negative pathogen evaluation bears the risk of inadequate antibiotic treatment, while inconsistent pathogen identification may result in inappropriate antibiotic selection, both potentially leading to recurrent PJI. On the other hand, a false-positive microbial analysis can be associated with overtreatment and the occurrence of serious side effects caused by various classes of antibiotics.

To the best of our knowledge, no comparative study has highlighted potential inconsistencies in microbiological analyses in the context of septic revision arthroplasty. In a previous study, we evaluated intraoperative microbiological findings during aseptic revision surgery and demonstrated a significant variability between two different laboratories [7]. More precisely, the evaluation of 60 patients with 600 associated samples (300 per laboratory) revealed poor correlation between two certified laboratories in terms of unexpected positive intraoperative culture (UPIC) results (k = 0.162). Consistent microbial findings between the respective laboratories were only observed in one (1.6%) patient [7]. Compared to our previous work, we found a higher degree of consistency between laboratories in the course of septic revision arthroplasty. Nevertheless, the present study may entail critical clinical implications. Taken together, in 5 of 40 patients (12.5%), significant discrepancies in microbiological analyses were found, which in turn required a different antibiotic treatment. In these cases, the additional interpretation of the other laboratory’s results necessitated a therapeutic response. The observed discrepancies pose a substantial challenge for postoperative decision-making. This, in turn, emphasizes the need for cautious interpretation.

This study has some limitations. First, our analysis was limited by the small number of septic revision surgeries due to the single-center study design. Advanced techniques such as sonication or next-generation sequencing were not performed routinely. Additionally, microbial contamination during sample collection cannot be excluded entirely. However, it is worth mentioning that all samples were obtained under strict sterile conditions by the performing surgeon, as per a standardized clinical protocol. Lastly, it has to be mentioned that there was a difference of 4 days regarding sample incubation time among laboratories, leading to a potential source of variability. The ideal incubation period for microbiological analysis of acute and chronic periprosthetic infections is still subject to debate. In a recent study, Morreel et al. considered 7 days of incubation to be sufficient in order to diagnose acute infections. In chronic cases, where pathogen growth can be delayed, 94% of infections were detected within 10 days of incubation, whereas only 5 of 67 cases exhibited growth after day 10 [15]. Birlutiu et al. claimed to diagnose 100% of acute and chronic PJIs within 8 days of incubation [16]. However, due to the fact that the longest duration to pathogen detection took 8 days of incubation, the authors believe that the different time spans did not affect the observed results of the present study.

This is the first study which analyses the concordance of microbiological results in septic revision arthroplasty. It is worth noting that even certified institutions exhibit considerable discrepancies. Given the fact that microbial evaluation is inherently prone to any kind of impurities, the observed differences may be partly attributed to unintended bacterial contamination during sample processing. Other differences may be a consequence of polymicrobial infections, which were presumably also detected due to the high number of specimens analyzed per patient.

Taken together, interlaboratory variability remains a significant barrier to reliable PJI diagnosis, with direct therapeutic implications in 12.5% of cases. Standardized microbiological protocols and confirmatory testing strategies are urgently needed.

## Figures and Tables

**Table 1 jcm-14-05582-t001:** Divergent microbiological results according to respective laboratory analyses.

Patient No.	Sex	Age (y)	Revision TKA/THA	Pathogen Spectrum Laboratory 1	Pathogen Spectrum Laboratory 2	Preoperative Joint Aspiration			Histology
						Pathogen	WBC Count	PMN (%)	M & K Type
1	m	71.3	THA		*Strept. dysgalactiae* (5)	*Strept. Dysgalactiae*	57,400	88	2
2	f	77.4	TKA	*E. faecalis* (1, 2, 5) *Staph. sciuri* (3) *Kocuria rosea* (3) *Kocuria kristinae* (4) *Staph. epidermidis* (4, 5)	*E. faecalis* (1–4) *Staph. epidermidis* (3) *Corynebact. tuberculostearicum* (3)	*Staph. Haemolyticus*, *Kocuria rosea*	n.a.	n.a.	1
3	m	66.5	TKA	*E. faecalis* (2) *Staph. epidermidis* (2) *Staph. capitis* (2, 3, 4, 5)	*Staph. epidermidis* (1) *Staph. capitis* (2–5)	*Staph. Capitis*	1220	80	3
4	f	56.8	TKA	*Staph.caprae* (1, 3, 4, 5) *Staph. lugdunensis* (2,5) *Aerococcus viridans* (2, c)	*Staph. caprae* (1–5)	*Staph. caprae*	49,400	73	2
5	f	76.2	THA	*Acinetobacter baumanii* complex (3) *Staphylococcus capitis* (5)	*Cutibacterium acnes* (1, 2, 3, 5)	*Cutibacterium acnes*	1200	9	2
6	m	57.9	THA	*Rhizibuim radiobacter* (2, c) *Cutibacterium acnes* (4)	*Staph. saccharolyticus* (1, c) (2, 3)	*Staph. epidermidis*, *Cutibacterium acnes*	n.a.	n.a.	2
7	f	69.3	THA	*Staph. epidermidis* (1, 2, 3, 5) *Staph. lentus* (5, c) *Leuconostoc mesenteroides* (5, c)	*Staph. epidermidis* (1, 2, 3, 5)	*Staph. epidermidis*	24,000	73	2
8	f	72.3	TKA	*Staph. aureus* (1–5) *Staph. hominis* ssp. Hominis (5, c)	*Staph. aureus* (1–5)		1200	92	2
9	m	68.3	THA	*Staph. capitis* (2) *Staph. saccharolyticus* (4) *Staph. hominis* ssp. Hominis (5)	*Staph. epidermidis* (4, 5)		n.a.	n.a.	2
10	f	83.7	THA	*Staph. pseudintermedius* (1, 3, 4, 5) *Staph. lugdunensis* (2, 4, 5)	*Staph. lugdunesis* (1–5)		150,000	98	1
11	f	78.7	THA	*Staph. hominis* ssp. Hominis (2, 3) *Staph. epidermidis* (4, 5)	*Staph. epidermidis* (2–5)	*Staph. epidermidis*	34,000	81	3
12	m	76.3	THA	*Pseudomonas aeruginosa* (1–5) *Pseudomonas fluorescens* (4) *Staph. epidermidis* (3, c)	*Pseudomonas aeruginosa* (1–5) *Bacillus* species (3, c)	*Pseudomonas aeruginosa*	n.a.	n.a.	n.a.
13	m	81.1	TKA	*Peptoniphilus asaccharolyticus* (3)			35,400	86	3
14	f	79.6	TKA	*Staph. pseudintermedius* (1–5)	*Staph. aureus* (1–5)	*Staph. aureus*	n.a.	n.a.	3
15	f	87.3	THA		*Staph. epidermidis* (1, 2)	*Staph. epidermidis*	24,700	37	3

no.: number; y: years; WBC: white blood cell; PMN: polymorphonuclear leukocyte percentage; n.a.: cell count not available due to insufficient amount of joint fluid aspiration or sample clotting. Numbers in brackets indicate the respective microbiological sample number. c: contamination. Histological evaluation was conducted according to Morawietz and Krenn (M and K) classification [9].

## Data Availability

Data is contained within the article (the original contributions presented in this study are included in the article; further inquiries can be directed to the corresponding author).

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
