# Peer review of "Prospective Comparative Analysis of Simultaneous Microbiological Assessment in Septic Revision Arthroplasty: Can We Rely on Standard Diagnostics?"

_jcm, 2025, doi:10.3390/jcm14155582_

Round 1

Reviewer 1 Report

Comments and Suggestions for Authors

Thank you for submitting your manuscript titled “Reliability of Simultaneous Microbiological Assessment in Septic Revision Arthroplasty.”
The manuscript investigates the reliability of standard microbiological workflows in the setting of septic revision arthroplasty, specifically when samples are processed independently by two accredited laboratories. This topic is clinically relevant due to the central role of pathogen identification in determining antibiotic management and surgical planning during revision procedures. The study is based on a prospective design with predefined inclusion criteria aligned with EBJIS recommendations and follows a standardized protocol for sample acquisition in an operative environment under controlled conditions. The authors also reference previous work on aseptic revisions, which provides context and allows for comparison across settings.

The methodology is generally consistent with current clinical practice, incorporating five microbiological samples per patient, histopathological classification using the Morawietz and Krenn system, and excluding patients with recent antibiotic use or immunosuppression. However, certain elements require clarification. The definition of a “significantly positive” result is based on pathogen virulence and the number of positive samples, but the approach to discordant polymicrobial findings is not fully addressed. Further detail on how discrepancies in polymicrobial cultures were interpreted, particularly in cases where antibiotic regimens were adjusted, would improve transparency.

The use of Cohen’s kappa coefficient to assess interlaboratory agreement is methodologically appropriate. However, the interpretation of the reported kappa value (0.227) as “fair” may not fully reflect the clinical implications of such diagnostic discordance. Although the majority of discrepancies did not alter therapeutic plans, the subset of cases in which antimicrobial therapy was modified based on laboratory-specific findings warrants further analysis. It would be useful to clarify whether these treatment adaptations influenced clinical outcomes or follow-up results.

The discussion adequately situates the findings within the context of microbiological diagnostic variability, though further elaboration on procedural and technical differences between laboratories would strengthen the manuscript. The discrepancy in incubation periods (10 versus 14 days) is mentioned but not explored in depth as a potential source of variability. Additionally, the potential influence of fastidious or biofilm-associated organisms is not addressed. The possibility of contamination is noted, but the criteria used to identify or exclude contamination in discrepant cases are not clearly described.

The limitations are appropriately acknowledged, including the single-center setting, the relatively small number of cases, and the absence of adjunct diagnostic methods such as sonication or molecular testing. These constraints may affect the generalizability of the findings, particularly in institutions employing broader diagnostic panels. Nonetheless, the study addresses a diagnostic question that remains unresolved in current literature.

The manuscript is written clearly and follows a logical structure. However, some aspects, particularly in the presentation of statistical results and therapeutic implications, could benefit from greater specificity. For instance, the phrase “a more extensive antibiotic treatment” would be clearer if defined in terms of antimicrobial spectrum or treatment duration. Similarly, the descriptors used to qualify levels of agreement (e.g., “fair”) should refer to recognized statistical thresholds.

The study presents data on interlaboratory variability in microbiological analysis of septic arthroplasty revisions and illustrates how such discrepancies may influence therapeutic decisions in a subset of patients. The findings support the need for integrating clinical, histological, and microbiological information when managing periprosthetic joint infection. Minor revisions are necessary to clarify aspects of methodology, elaborate on the interpretation of discordant findings, and refine the discussion of laboratory variability. Following these adjustments, the manuscript would be suitable for consideration.Kind regards.

Author Response

Point-by-point response to the comments of Reviewer 1

Title:

Old title: Reliability of simultaneous microbiological assessment in septic revision arthroplasty

New title: Prospective comparative analysis of simultaneous microbiological assessment in septic revision arthroplasty: Can we rely on standard diagnostics?

Journal: Journal of Clinical Medicine

Manuscript ID: jcm-3786772

Status: pending major revisions

Review of Reviewer 1:

Thank you for submitting your manuscript titled “Reliability of Simultaneous Microbiological Assessment in Septic Revision Arthroplasty.”
The manuscript investigates the reliability of standard microbiological workflows in the setting of septic revision arthroplasty, specifically when samples are processed independently by two accredited laboratories. This topic is clinically relevant due to the central role of pathogen identification in determining antibiotic management and surgical planning during revision procedures. The study is based on a prospective design with predefined inclusion criteria aligned with EBJIS recommendations and follows a standardized protocol for sample acquisition in an operative environment under controlled conditions. The authors also reference previous work on aseptic revisions, which provides context and allows for comparison across settings.

The methodology is generally consistent with current clinical practice, incorporating five microbiological samples per patient, histopathological classification using the Morawietz and Krenn system, and excluding patients with recent antibiotic use or immunosuppression.

Specific comments:

  • Reviewer 1: However, certain elements require clarification. The definition of a “significantly positive” result is based on pathogen virulence and the number of positive samples, but the approach to discordant polymicrobial findings is not fully addressed. Further detail on how discrepancies in polymicrobial cultures were interpreted, particularly in cases where antibiotic regimens were adjusted, would improve transparency.

Response: Thank you for mentioning this important aspect. We specified the rating of bacterial results according to the EBJIS classification (line 148-156: “Microbiological results were classified as follows: Two samples with growth of the same pathogen or detection of a highly virulent pathogen in one sample were classified as periprosthetic infection. Pathogen growth in a single sample was carefully analyzed according to EBJIS-specific definitions (accordance with preoperative joint aspiration or presence of common contaminants): Intraoperative samples generating a single moderate growth of a low-virulent pathogen were classified as contamination if preoperative joint aspiration remained sterile. In the presence of single moderate growth of a low-virulent pathogen and a simultaneous growth of the same pathogen during preoperative joint aspiration, PJI was confirmed. ”).

Furthermore, we added information on treatment in those cases with an adjusted antibiotic therapy (line 251-256: “In 2 patients (patients 5 and 10), the diverging results led to a more extensive antibiotic treatment compared to the result interpretation of one institution. In patient 5, the detection of Acinetobacter baumanii resulted in additional treatment with Trimethoprim/sulfamethoxazole compared to an isolated treatment with amoxicilline for Cutiacterium acnes. In patient 10, the antibiograms of the detected pathogens differed as Staphylococcus lugdunensis was oxacillin-susceptible while Staphylococcus pseudintermedius was declared oxacillin-resistant. For that reason, antibiotic therapy with fosfomycin was extended by adding daptomycin.”).

  • Reviewer 1: The use of Cohen’s kappa coefficient to assess interlaboratory agreement is methodologically appropriate. However, the interpretation of the reported kappa value (0.227) as “fair” may not fully reflect the clinical implications of such diagnostic discordance. Although the majority of discrepancies did not alter therapeutic plans, the subset of cases in which antimicrobial therapy was modified based on laboratory-specific findings warrants further analysis. It would be useful to clarify whether these treatment adaptations influenced clinical outcomes or follow-up results.

Response: Thank you for this cause for thought. In line with our investigations, the definition of cohen’s kappa can be declared as “fair” or “slight”. According to the interpretation of Koch and Landis (1977), k=0.227 is rated and defined as “slight”. We furthermore added the 95% confidence interval for Cohen’s kappa. Given your considerations, we changed the respective wording and the manuscript was adjusted accordingly (line 28-29: “Kappa correlation coefficient indicated a slight value in terms of data consistency between institutions (k = 0.227, p = 0.151).” ; line 223-224: “Kappa correlation coefficient indicated a slight value in terms of data consistency between institutions (k=0.227, p=0.151, 95% confidence interval [-0.085, 0.539]).” ; line 298-301: “The main finding of this study reveals a slight and relatively low correlation between the observed institutions in terms of data consistency (k = 0.227, p = 0.151). More precisely, this means that while there's some agreement, it's not strong and is only slightly better than what might be expected by chance.”).

The effect on the antibiotic treatment in cases with discrepancies has been added and the manuscript was changed accordingly as mentioned above (line 249-256: In 2 patients (patients 5 and 10), the diverging results led to a more extensive antibiotic treatment compared to the result interpretation of one institution. In patient 5, the detection of Acinetobacter baumanii resulted in additional treatment with Trimethoprim/sulfamethoxazole compared to an isolated treatment with amoxicilline for Cutiacterium acnes. In patient 10, the antibiograms of the detected pathogens differed as Staphylococcus lugdunensis was oxacillin-susceptible while Staphylococcus pseudintermedius was declared oxacillin-resistant. For that reason, antibiotic therapy with fosfomycin was extended by adding daptomycin.”)

Regarding the clinical influence on middle-and long-term follow-up, the authors state that this aspect was not a part of the outcome evaluation, as we primarily aimed to analyze potential deviations with regard to microbiological sample workup. However, we clearly see the reviewer’s point, which aims to unravel the clinical implication of the observed findings. Against this background, the patients of this study will further be investigated in our outpatient department every 12 months in order to receive a concise follow-up and delineate the implications for middle- and long-term follow-up.

  • The discussion adequately situates the findings within the context of microbiological diagnostic variability, though further elaboration on procedural and technical differences between laboratories would strengthen the manuscript. The discrepancy in incubation periods (10 versus 14 days) is mentioned but not explored in depth as a potential source of variability.

Response: Thank you for your comment. With respect to the different incubation times, we agree on the potential impact on pathogen detection. The manuscript was changed accordingly (line 363-373: “Lastly, it has to be mentioned that there was a difference of 4 days regarding sample incubation time among laboratories, leading to a potential source of variability. The ideal incubation period for microbiological analysis of acute and chronic periprosthetic infections is still subject of debate. In a recent study, Morreel et al. considered 7 days of incubation to be sufficient in order to diagnose acute infections. In chronic cases, where pathogen growth can be delayed, 94% of infections were detected within 10 days of incubation, whereas only 5 of 67 cases exhibited growth after day 10 [13]. Birlutiu et al. claimed to diagnose 100% of acute and chronic PJIs within 8 days of incubation [14]. However and due to the fact that the longest duration till pathogen detection took 8 days of incubation, the authors believe that the different time spans did not affect the observed results of the present study.“). Taken together, we believe that incubation time discrepancy did not significantly influence our results. Nevertheless, we agree with reviewer 1 that different time spans may evoke a potential source of variability.

  • Additionally, the potential influence of fastidious or biofilm-associated organisms is not addressed. The possibility of contamination is noted, but the criteria used to identify or exclude contamination in discrepant cases are not clearly described.

Response: Thank you for your comment. We are aware of the fact that the interpretation of positive results, especially in the context of discrepant findings represents a substantial challenge. According to the stated limitations of our study (line 358-373), removed implants did not undergo sonication on a routine basis. Thus, the detection of biofilm-associated germs may be underrepresented. However, sonication also poses a certain thread for contamination during sample workup and sample transport. Due to the primary aim of this study, we therefore solely focused on tissue samples. The criteria for contamination were redefined in the manuscript (line 147-156: “Microbiological results were classified as follows: Two samples with growth of the same pathogen or detection of a highly virulent pathogen in one sample were classified as periprosthetic infection. Pathogen growth in a single sample was carefully analyzed according to EBJIS-specific definitions (accordance with preoperative joint aspiration or presence of common contaminants): Intraoperative samples generating a single moderate growth of a low-virulent pathogen were classified as contamination if preoperative joint aspiration remained sterile. In the presence of single moderate growth of a low-virulent pathogen and a simultaneous growth of the same pathogen during preoperative joint aspiration, PJI was confirmed.“).

  • The limitations are appropriately acknowledged, including the single-center setting, the relatively small number of cases, and the absence of adjunct diagnostic methods such as sonication or molecular testing. These constraints may affect the generalizability of the findings, particularly in institutions employing broader diagnostic panels. Nonetheless, the study addresses a diagnostic question that remains unresolved in current literature.

Response: Thank you for your meticulous analysis. We are aware of the limitations of our study. However, we are happy to meet the reviewer’s expectations with regard to the currently unsolved diagnostic questions.

  • The manuscript is written clearly and follows a logical structure. However, some aspects, particularly in the presentation of statistical results and therapeutic implications, could benefit from greater specificity. For instance, the phrase “a more extensive antibiotic treatment” would be clearer if defined in terms of antimicrobial spectrum or treatment duration. Similarly, the descriptors used to qualify levels of agreement (e.g., “fair”) should refer to recognized statistical thresholds.

Response: Thank you again for your fruitful comments. According to the reviewer’s suggestions, we specified antibiotic treatment as well as the rating regarding the respective levels of agreement as stated above. (line 251-256: In 2 patients (patients 5 and 10), the diverging results led to a more extensive antibiotic treatment compared to the result interpretation of one institution. In patient 5, the detection of Acinetobacter baumanii resulted in additional treatment with Trimethoprim/ sulfamethoxazole compared to an isolated treatment with amoxicilline for Cutiacterium acnes. In patient 10, the antibiograms of the detected pathogens differed as Staphylococcus lugdunensis was oxacillin-susceptible while Staphylococcus pseudintermedius was declared oxacillin-resistant. For that reason, antibiotic therapy with fosfomycin was extended by adding daptomycin.” ; line 28-29: “Kappa correlation coefficient indicated a slight value in terms of data consistency between institutions (k = 0.227, p = 0.151).” ; line 223-224: “Kappa correlation coefficient indicated a slight value in terms of data consistency between institutions (k=0.227, p=0.151, 95% confidence interval [-0.085, 0.539]).” ; line 298-301: “The main finding of this study reveals a slight and relatively low correlation between the observed institutions in terms of data consistency (k = 0.227, p = 0.151). More precisely, this means that while there's some agreement, it's not strong and is only slightly better than what might be expected by chance.”).

  • The study presents data on interlaboratory variability in microbiological analysis of septic arthroplasty revisions and illustrates how such discrepancies may influence therapeutic decisions in a subset of patients. The findings support the need for integrating clinical, histological, and microbiological information when managing periprosthetic joint infection. Minor revisions are necessary to clarify aspects of methodology, elaborate on the interpretation of discordant findings, and refine the discussion of laboratory variability. Following these adjustments, the manuscript would be suitable for consideration. Kind regards.

Response: We are pleased to notice that our study seems to contain some valuable information in the reviewer’s eyes. Once more, we’d like to thank you for raising some critical issues, which helped us further to improve our manuscript.

Reviewer 2 Report

Comments and Suggestions for Authors

Title and Abstract
The title reflects the central topic but lacks methodological precision. Including the study design (e.g., “prospective comparative analysis of dual-laboratory microbiological diagnostics in septic revision arthroplasty”) would improve indexing and clarity regarding the level of evidence.

The abstract is overly generic and insufficiently structured. Key findings, such as the number of patients with discordant results (15 of 40, 37.5%) and the clinical impact (12.5% requiring antibiotic modification), should be explicitly stated. The conclusion should avoid vague phrasing like “pose a significant challenge” and instead emphasize the implications for therapeutic decision-making and the need for standardized microbiological protocols.

Introduction
The introduction emphasizes the importance of microbial analysis in septic revision arthroplasty but is verbose and lacks a sharp, hypothesis-driven narrative. Several sentences (e.g., “paves the way for successful treatment of periprosthetic joint infections”) are redundant and should be condensed.

The authors should deepen the discussion of interlaboratory variability by referencing diagnostic standards such as EBJIS and IDSA criteria, while highlighting the clinical risks of inconsistent pathogen identification (e.g., overtreatment or inappropriate antibiotic selection). Additionally, modern diagnostic techniques—such as sonication and molecular assays—should be mentioned to frame the study’s clinical relevance against contemporary standards.

Methods
The study design is prospective and generally well described. However, several key aspects require clarification: Sample handling and contamination controls: The rationale for sending samples to two different labs is clear, but no mention is made of negative controls or environmental contamination checks. Definition of contamination: The criteria used (e.g., “single moderate growth of a low-virulent pathogen”) are arbitrary and not referenced to established guidelines. This must be aligned with EBJIS or MSIS criteria. Laboratory protocol differences: The use of distinct incubation times (14 vs. 10 days) introduces bias and requires a detailed justification. The potential impact on pathogen detection should be acknowledged in both Methods and Discussion. Statistics: Cohen’s kappa is reported, but 95% confidence intervals are not provided. No power calculation is mentioned, which is critical given the small sample size (n=40).

Results
The results are presented clearly but lack clinical interpretation in several areas: The kappa value (0.227) indicates poor to fair agreement and should be discussed more critically, particularly regarding its implications for patient care. The clinical consequences of discordant findings are underdeveloped. A detailed table specifying which pathogens were inconsistently identified, and how antibiotic regimens were adjusted, would add significant value. Reporting of sterile cases diagnosed solely on histology or clinical suspicion (e.g., “type II membrane with normal cell count”) should be contextualized against established PJI diagnostic algorithms.

Discussion
The discussion largely reiterates the results rather than critically interpreting them. Key areas for improvement include: Clinical impact: The authors should explore how diagnostic discrepancies could result in overtreatment, undertreatment, or delayed infection eradication. Comparison to literature: The findings should be contrasted with published studies on laboratory variability in PJI diagnostics, including modern techniques such as prolonged cultures, PCR, and sonication. Bias analysis: Differences in incubation protocols and culture media between the two laboratories must be critically assessed as potential sources of variability. Limitations: The small sample size, single-center design, and absence of advanced microbial techniques (e.g., NGS) need explicit emphasis.

Conclusion
The conclusion is too general and underplays the clinical consequences of laboratory discordance. A stronger statement would be: “This study demonstrates that interlaboratory variability remains a significant barrier to reliable PJI diagnosis, with direct therapeutic implications in 12.5% of cases. Standardized protocols and confirmatory testing strategies are urgently needed.”

Comments on the Quality of English Language

Language and Style
The manuscript is readable but requires polishing: Redundancies should be eliminated (e.g., “tissue samples for microbiological analysis… microbiological workup”). Terminology consistency: Ensure uniform use of abbreviations (e.g., PJI, THA, TKA). Active voice should be preferred where possible: e.g., “Samples were collected” → “We collected samples.”

Author Response

Point-by-point response to the comments of Reviewer 2

Title:

Old title: Reliability of simultaneous microbiological assessment in septic revision arthroplasty

New title: Prospective comparative analysis of simultaneous microbiological assessment in septic revision arthroplasty: Can we rely on standard diagnostics?

Journal: Journal of Clinical Medicine

Manuscript ID: jcm-3786772

Status: pending major revisions

Review of Reviewer 2:

Specific comments:

  • Reviewer 2: Title and Abstract        
    The title reflects the central topic but lacks methodological precision. Including the study design (e.g., “prospective comparative analysis of dual-laboratory microbiological diagnostics in septic revision arthroplasty”) would improve indexing and clarity regarding the level of evidence.

Response: Thank you for your comment and the valuable suggestion. According to your opinion, the title has been rephrased and the manuscript was changed accordingly (“Prospective comparative analysis of simultaneous microbiological assessment in septic revision arthroplasty: Can we rely on standard diagnostics?”)

  • Reviewer 2: The abstract is overly generic and insufficiently structured. Key findings, such as the number of patients with discordant results (15 of 40, 37.5%) and the clinical impact (12.5% requiring antibiotic modification), should be explicitly stated. The conclusion should avoid vague phrasing like “pose a significant challenge” and instead emphasize the implications for therapeutic decision-making and the need for standardized microbiological protocols.

Response: We clearly see the reviewer’s point and thank you for this comment. According to your suggestions, essential key findings of this study were explicitly stated and the manuscript was structured and changed accordingly. The wording was adjusted in to avoid vague phrasing. (line 21-34: “Results: Each laboratory analyzed 200 samples. Twenty-five patients (62.5%) showed consistent results between laboratories. Diverging results were observed in 15 of 40 patients (37.5%). Of these, 3 individuals showed pathogen growth in only 1 laboratory. In 12 patients with discrepant results, laboratory analyses revealed a partly different pathogen spectrum. With regard to clinical impact and infection eradication, the respective differences implicated a therapeutic response by a change of the administered postoperative antibiotic treatment in 5 (12.5%) of patients. Kappa correlation coefficient indicated a slight value in terms of data consistency between institutions (k=0.227, p=0.151). Conclusions: The majority of evaluated samples show comparable results with regard to microbiological evaluation. Nevertheless, a substantial number of specimens were classified differently. The observed discrepancies pose a challenge for postoperative decision-making. Against this background, standardized microbiological protocols remain mandatory for a conclusive clinical implication to eradicate PJI.”)

  • Introduction
    The introduction emphasizes the importance of microbial analysis in septic revision arthroplasty but is verbose and lacks a sharp, hypothesis-driven narrative. Several sentences (e.g., “paves the way for successful treatment of periprosthetic joint infections”) are redundant and should be condensed.

Response: Thank you for this valuable comment. The Introduction has been adjusted according to your suggestions by removing verbose wording. Redundant information has been deleted and the manuscript was changed accordingly (line 78-84: “Due to the inherent question of diagnostic reliability, the primary aim of this study was to analyze microbiological findings from septic revision hip and knee arthroplasty procedures and to compare the diagnostic accuracy between two internationally certified and accredited laboratories. We hypothesized that in septic situations, interlaboratory results would differ in terms of pathogen identification rate and bacterial species classification. Consequently, the purpose of this study was to estimate the prevalence of diverging laboratory results as well as to point out their clinical implications.“).

  • The authors should deepen the discussion of interlaboratory variability by referencing diagnostic standards such as EBJIS and IDSA criteria, while highlighting the clinical risks of inconsistent pathogen identification (e.g., overtreatment or inappropriate antibiotic selection). Additionally, modern diagnostic techniques—such as sonication and molecular assays—should be mentioned to frame the study’s clinical relevance against contemporary standards.

Response: Thank you for your comment and the associated suggestions. According to the stated limitations of our study (line 358-373), removed implants did not undergo sonication on a routine basis. Thus, the detection of biofilm-associated germs may be underrepresented. However, sonication also poses a certain thread for contamination during sample workup and sample transport. Due to the primary aim of this study, we therefore solely focused on tissue samples. With regard to the clinical risks of inconsistent pathogen identification, we hope to meet the reviewer’s expectations and adjusted the discussion of the manuscript, which has been changed accordingly (line 323-328: The consequences of incorrect pathogen identification may imply significant patient specific complications: False-negative pathogen evaluation bears the risk of inadequate antibiotic treatment, while inconsistent pathogen identification may result in inappropriate antibiotic selection, both potentially leading to recurrent PJI. On the other hand, a false-positive microbial analysis can be associated with overtreatment and the occurrence of serious side effects caused by various classes of antibiotics.“) ; line 287-292: In this context, diagnostic standards such as those of the European Bone and Joint Infection Society (EBJIS) and the Infectious Diseases Society of America (IDSA) emphasize the importance of pathogen concordance and quantitative thresholds in defining PJI [5,10]. Yet, these definitions rely heavily on the accuracy of microbiological culture results, which, as our study demonstrates, can be undermined by interlaboratory variability—even when protocols are standardized.).

  • Methods
    The study design is prospective and generally well described. However, several key aspects require clarification: Sample handling and contamination controls: The rationale for sending samples to two different labs is clear, but no mention is made of negative controls or environmental contamination checks. Definition of contamination: The criteria used (e.g., “single moderate growth of a low-virulent pathogen”) are arbitrary and not referenced to established guidelines. This must be aligned with EBJIS or MSIS criteria. Laboratory protocol differences: The use of distinct incubation times (14 vs. 10 days) introduces bias and requires a detailed justification. The potential impact on pathogen detection should be acknowledged in both Methods and Discussion. Statistics: Cohen’s kappa is reported, but 95% confidence intervals are not provided. No power calculation is mentioned, which is critical given the small sample size (n=40).

Response: Thank you for your valuable comment. The laboratories are accredited and certified according to international standards (DIN EN ISO 15189). Environmental contamination checks and negative controls are performed in the laboratories on a routine basis and were not part of the present study. We specified the rating of bacterial results according to the EBJIS classification. The manuscript was changed accordingly (line 147-156: “Microbiological results were classified as follows: Two samples with growth of the same pathogen or detection of a highly virulent pathogen in one sample were classified as periprosthetic infection. Pathogen growth in a single sample was carefully analyzed according to EBJIS-specific definitions (accordance with preoperative joint aspiration or presence of common contaminants): Intraoperative samples generating a single moderate growth of a low-virulent pathogen were classified as contamination if preoperative joint aspiration remained sterile. In the presence of single moderate growth of a low-virulent pathogen and a simultaneous growth of the same pathogen during preoperative joint aspiration, PJI was confirmed.“).                                                                                       With regard to the missing power analysis, we clearly see the reviewer’s point and appreciate the straight message. According to the stated methods section, we aimed for a homogenous patient cohort, all suffering from PJI. This in turn was associated with a small sample size due to the strict exclusion criteria.

With respect to the different incubation times, we agree on the potential impact on pathogen detection. The manuscript was changed accordingly (line 363-373 : “Lastly, it has to be mentioned that there was a difference of 4 days regarding sample incubation time among laboratories, leading to a potential source of variability. The ideal incubation period for microbiological analysis of acute and chronic periprosthetic infections is still subject of debate. In a recent study, Morreel et al. considered 7 days of incubation to be sufficient in order to diagnose acute infections. In chronic cases, where pathogen growth can be delayed, 94% of infections were detected within 10 days of incubation, whereas only 5 of 67 cases exhibited growth after day 10 [13]. Birlutiu et al. claimed to diagnose 100% of acute and chronic PJIs within 8 days of incubation [14]. However and due to the fact that the longest duration till pathogen detection took 8 days of incubation, the authors believe that the different time spans did not affect the observed results of the present study.“). Taken together, we believe that incubation time discrepancy did not significantly influence our results. Nevertheless, we agree with reviewer 1 that different time spans may evoke a potential source of variability.

According to the reviewer’s suggestion, 95% confidence interval regarding Cohen’s kappa was provided in the manuscript:

(line 223-224: “Kappa correlation coefficient indicated a slight value in terms of data consistency between institutions (k=0.227, p=0.151, 95% confidence interval [-0.085, 0.539]).

  • Results
    The results are presented clearly but lack clinical interpretation in several areas: The kappa value (0.227) indicates poor to fair agreement and should be discussed more critically, particularly regarding its implications for patient care.

Response: Thank you for this cause for thought. In line with our investigations, the definition of cohen’s kappa can be declared as “fair” or “slight”. According to the interpretation of Koch and Landis (1977), k=0.227 is rated and defined as “slight”. Given your considerations, we changed the respective wording and the manuscript was adapted accordingly (line 223-224 (as stated above); line 298-301: “The main finding of this study reveals a slight and relatively low correlation between the observed institutions in terms of data consistency (k = 0.227, p = 0.151). More precisely, this means that while there's some agreement, it's not strong and is only slightly better than what might be expected by chance.”)).

With regard to the clinical implications, we stated this more clearly in the manuscript (line 323-328: “The consequences of incorrect pathogen identification may imply significant patient specific complications: False-negative pathogen evaluation bears the risk of inadequate antibiotic treatment, while inconsistent pathogen identification may result in inappropriate antibiotic selection, both potentially leading to recurrent PJI. On the other hand, a false-positive microbial analysis can be associated with overtreatment and the occurrence of serious side effects caused by various classes of antibiotics.”)

  • The clinical consequences of discordant findings are underdeveloped. A detailed table specifying which pathogens were inconsistently identified, and how antibiotic regimens were adjusted, would add significant value. Reporting of sterile cases diagnosed solely on histology or clinical suspicion (e.g., “type II membrane with normal cell count”) should be contextualized against established PJI diagnostic algorithms.

Response: Thank you for this comment. We see your point and the raised concerns. Consequently, we added information on treatment in those cases with adjustment of the antibiotic therapy (line 251-256: In 2 patients (patients 5 and 10), the diverging results led to a more extensive antibiotic treatment compared to the result interpretation of one institution. In patient 5, the detection of Acinetobacter baumanii resulted in additional treatment with Trimethoprim/sulfamethoxazole compared to an isolated treatment with amoxicilline for Cutiacterium acnes. In patient 10, the antibiograms of the detected pathogens differed as Staphylococcus lugdunensis was oxacillin-susceptible while Staphylococcus pseudintermedius was declared oxacillin-resistant. For that reason, antibiotic therapy with fosfomycin was extended by adding daptomycin.“). We furthermore believe, that relevant information whether pathogen findings were inconsistent or concordant between the two laboratories, is provided in Table 1, as the results of the two laboratories are displayed for each case and contaminations are marked as such. Regarding the cases of culture-negative PJI, the respective histological analysis was taken into account according to the EBJIS criteria. In their publication, Mc Nally et al. stated that the presence of ≥ five neutrophils in ≥ five high power fields (corresponding to an infectious membrane according to Morawietz and Krenn type 2 and 3) goes in line with a confirmed PJI (McNally et al., BJJ, 2021).

  • Discussion
    The discussion largely reiterates the results rather than critically interpreting them. Key areas for improvement include: Clinical impact: The authors should explore how diagnostic discrepancies could result in overtreatment, undertreatment, or delayed infection eradication. Comparison to literature: The findings should be contrasted with published studies on laboratory variability in PJI diagnostics, including modern techniques such as prolonged cultures, PCR, and sonication. Bias analysis: Differences in incubation protocols and culture media between the two laboratories must be critically assessed as potential sources of variability. Limitations: The small sample size, single-center design, and absence of advanced microbial techniques (e.g., NGS) need explicit emphasis.

Response: Thank you for your valuable comment. With regard to the clinical impact of the observed findings, we added some thoughts dealing with the consequences of discrepant diagnostic results (line 323-328: The consequences of incorrect pathogen identification may imply significant patient specific complications: False-negative pathogen evaluation bears the risk of inadequate antibiotic treatment, while inconsistent pathogen identification may result in inappropriate antibiotic selection, both potentially leading to recurrent PJI. On the other hand, a false-positive microbial analysis can be associated with overtreatment and the occurrence of serious side effects caused by various classes of antibiotics.“). Furthermore, differences in incubation protocols between the two laboratories were discussed and the manuscript has been changed accordingly as stated above (line 363-373: “Lastly, it has to be mentioned that there was a difference of 4 days regarding sample incubation time among laboratories, leading to a potential source of variability. The ideal incubation period for microbiological analysis of acute and chronic periprosthetic infections is still subject of debate. In a recent study, Morreel et al. considered 7 days of incubation to be sufficient in order to diagnose acute infections. In chronic cases, where pathogen growth can be delayed, 94% of infections were detected within 10 days of incubation, whereas only 5 of 67 cases exhibited growth after day 10 [13]. Birlutiu et al. claimed to diagnose 100% of acute and chronic PJIs within 8 days of incubation [14]. However and due to the fact that the longest duration till pathogen detection took 8 days of incubation, the authors believe that the different time spans did not affect the observed results of the present study.“). Taken together, we believe that incubation time discrepancy did not significantly influence our results. Nevertheless, we agree with reviewer 1 that different time spans may evoke a potential source of variability.

With regard to the raised concerns dealing with the limitations of our study, the absence of advanced techniques such as NGS were already initially mentioned in our work (line 359-360: “Advanced techniques such as sonication or next-generation sequencing were not performed routinely.”).

The small sample size and single-center design were additionally discussed according to the reviewer’s suggestions (page xx, line xx: This study has some limitations. First, our analysis was limited by the small number of septic revision surgeries due to the single-center study design.”).

  • Conclusion
    The conclusion is too general and underplays the clinical consequences of laboratory discordance. A stronger statement would be: “This study demonstrates that interlaboratory variability remains a significant barrier to reliable PJI diagnosis, with direct therapeutic implications in 12.5% of cases. Standardized protocols and confirmatory testing strategies are urgently needed.”

Response: Thank you for this highly appreciated comment. We agree with the reviewer and thank you for your suggestion. The manuscript was changed accordingly (line 381-383: Taken together, interlaboratory variability remains a significant barrier to reliable PJI diagnosis, with direct therapeutic implications in 12.5% of cases. Standardized microbiological protocols and confirmatory testing strategies are urgently needed.”).

Finally, we are pleased to notice that our study seems to contain some valuable information in the reviewer’s eyes. Once more, we’d like to thank you for raising some critical issues, which helped us further to improve our manuscript.

Round 2

Reviewer 2 Report

Comments and Suggestions for Authors

I have carefully reviewed the authors’ detailed, point-by-point responses and the revised version of the manuscript. I appreciate the authors’ professional engagement with the critique and their substantial efforts to improve the scientific clarity, methodological transparency, and clinical interpretability of the work.

The revised manuscript reflects a clear enhancement in multiple dimensions: The title and abstract now accurately reflect the study design and emphasize the practical implications of the findings. The introduction is more concise, hypothesis-driven, and better contextualized within the framework of contemporary diagnostic standards. The methods section has been clarified to address critical aspects such as contamination definitions, incubation time discrepancies, and diagnostic classification criteria, with appropriate reference to EBJIS guidelines. In the results, the inclusion of clinical consequences and pathogen-specific details significantly increases the translational relevance. The discussion more effectively addresses the potential impact of interlaboratory variability on patient care and acknowledges the methodological limitations, including the absence of sonication and molecular techniques. The conclusion now offers a stronger, evidence-based statement on the need for standardized microbiological protocols. Moreover, the manuscript’s language and overall structure have improved, enhancing readability and scientific credibility.

Based on the modifications implemented in response to the initial review, I consider the revised manuscript to be significantly strengthened and now appropriate for publication, pending final editorial review. The study provides meaningful insights into the diagnostic variability of microbiological assessments in septic revision arthroplasty and contributes valuable evidence to the ongoing discourse on standardization in periprosthetic joint infection (PJI) diagnostics.

Best regards,

Prof. Dr. Sirbu Paul-Dan